

# Predictive value of plasma ephrinB2 levels for amputation risk following endovascular revascularization in peripheral artery disease

Pengcheng Guo[1,2], Lei Chen[3], Dafeng Yang[4], Lei Zhang[1,2], Chang Shu[1,2,5], Huande Li[3], Jieting Zhu[1,2], Jienan Zhou[1,2] and Xin Li[1,2]

[1] Vascular Surgery Department, the Secondary Xiangya Hospital, Central South University, Hunan, China
[2] Institute of Vascular Diseases, Central South University, Chang Sha, Hunan, China
[3] Pharmacy Department, the Secondary Xiangya Hospital, Central South University, Hunan, China
[4] Cardiology Surgery Department, the Secondary Xiangya Hospital, Central South University, Hunan, China
[5] State Key Laboratory of Cardiovascular Diseases, Center of Vascular Surgery, Fuwai Hospital National Center for Cardiovascular Diseases, Chinese Academy of Medical Science and Peking Union Medical College, Beijing, China

Corresponding author
Xin Li, lixin1981@csu.edu.cn

## ABSTRACT

**Background:** The aim of this study is to investigate the expression levels of ephrinB2 in patients with lower extremity peripheral arterial disease (PAD) and explore its association with the severity of the disease and the risk of amputation after endovascular revascularization.

**Methods:** During the period from March 2021 to March 2023, this study collected blood samples and clinical data from 133 patients diagnosed with lower extremity PAD and 51 healthy volunteer donors. The severity of lower extremity PAD patients was classified using the Rutherford categories. The expression of ephrin-B2 in plasma samples was detected using the Western Blotting.

**Results:** Compared to the control group, the levels of serum ephrinB2 in patients were significantly elevated ($p < 0.001$). Moreover, the plasma EphrinB2 levels were positively correlated with white blood cell counts (r = 0.204, $p = 0.018$), neutrophil counts (r = 0.174, $p = 0.045$), and neutrophil-to-lymphocyte ratio (NLR) (r = 0.223, $p = 0.009$). Furthermore, the AUCs of plasma ephrinB2 level, NLR, and their combination as predictors for amputation events within 30 months after lower extremity PAD endovascular revascularization were 0.659, 0.730 and 0.811. In the high-ephrinB2 group, the incidence of amputation events within 30 months after endovascular revascularization was higher.

**Conclusions:** Plasma EphrinB2 levels may be linked to lower extremity PAD development, inflammation, and postoperative amputation. Combining EphrinB2 and NLR can improve amputation prediction accuracy after endovascular revascularization in lower extremity PAD patients.

## INTRODUCTION

Peripheral arterial disease (PAD) of the lower extremities is a chronic disease primarily characterized by atherosclerosis, eventually leading to acute or chronic limb ischemia. More than 200 million people around the world suffer PAD, and the number is keep growing (*Lawall et al., 2016*). As the aging population continues to grow dramatically, PAD has been already emerged as a critical socioeconomic concern in China. The most common symptoms of the lower limb PAD is intermittent claudication, which is painful and harm of the walking ability severely. Moreover, symptoms like resting pain and ulcers on the foot/leg may start to manifest, potentially resulting in amputation of the affected limb (*Reinecke et al., 2015*). Furthermore, the development of PAD is often combined with carotid artery stenosis and coronary artery stenosis, exhibiting very higher risk of ischemic events and death compared to other cardiovascular diseases (*Saenz-Pipaon et al., 2021*).

There are large of scale studies demonstrated that the development of PAD is associated with atherosclerosis-related inflammation caused by pro-inflammatory cytokines and inflammatory signaling pathways (*Hansson & Hermansson, 2011*). Neutrophils have traditionally been regarded as bystanders or biomarkers of cardiovascular diseases (*Silvestre-Roig et al., 2020*), associated with innate immunity, while lymphocytes are typically involved in adaptive immune responses. Recent studies suggest that changes in the neutrophil-to-lymphocyte ratio (NLR) to some extent reflect the severity of inflammation in the body and are correlated with the prognosis of vascular reconstructive surgery (*Machado et al., 2018*).

The erythropoietin-producing human hepatocellular (Eph) receptors tyrosine kinases and their membrane-bound ligands, the ephrins, play a role not only in the communication signaling pathways between pericytes and endothelial cells but also in the inflammatory responses. EphrinB2, a member of the Eph receptor family, is thought to be involved in the regulation of inflammation in atherosclerosis (*Funk & Orr, 2013*; *Lehoux & Jones, 2016*). A previous study found that ephrinB2 expression was significantly increased in human atherosclerosis through gene expression analysis of atherosclerotic plaques in human carotid endarterectomy tissue (*Sakamoto et al., 2008*). Meanwhile, it was found that ephrinB2 expression elevates in the atherosclerotic endothelium of mouse aorta compared to non-atherosclerotic areas (*Braun et al., 2011*). In addition, in diabetes, an increase in the ephrin-B2 signaling pathway in peripapillary cells mediates retinal vascular inflammation (*Coucha et al., 2020*). However, the role of ephrinB2 related signaling in lower extremity lower extremity PAD remains unclear.

In our study, we investigated the expression levels of ephrinB2 in the plasma of lower extremity PAD patients. Additionally, we explored its ability to predict amputation in those post endovascular revascularization patients combined with clinical follow-up data.

## MATERIALS AND METHODS

### Study population

During the period from March 2021 to March 2023, we conducted this study at Xiangya Second Hospital of Central South University. To achieve 80% power and a one-sided

significance level of $\alpha = 0.05$, each of the healthy volunteer group and the PAD group requires at least 51 participants. A total number of 133 blood samples were collected from patients diagnosed with lower extremity PAD, and 51 blood samples were obtained from healthy volunteer donors.

The inclusion criteria for lower extremity PAD patients are as follows: (1) diagnosis by Doppler type B ultrasound or computed tomography angiogram (CTA), (2) clinical symptoms with Rutherford categories (*American College of Cardiology Foundation et al., 2011*; *Anderson et al., 2013*) between class 3 and class 6, (3) ABI ≤ 0.90 in either leg, (4) endovascular revascularization treatment was performed during the hospital stay. The exclusion criteria are: (1) severe immune system diseases or infectious diseases, (2) recent surgery or trauma, (3) incomplete clinical data, (4) Rutherford categories class 1 and 2. Healthy participants were enrolled if they had (1) no current or recent clinically significant somatic disease and (2) were currently taking no drugs.

This study was approved by the Ethics Committee of The Second Xiangya Hospital of Central South University (No. 2022-Clinical Research-113) and all participants in our study provided written informed consent.

## Data collection and clinical variables

Clinical covariates include age, gender, medical history, lipid abnormalities, liver and kidney function, inflammation markers, and blood routine, which were collected by reviewing and recording the clinical history system. The blood routine examination was conducted using an Automated Hematology Analyzer XN Series (XN-10, SYSMEX). Levels of liver function, renal function, and other biochemical indicators were measured using an Automatic Biochemistry Analyzer (7600-210; HITACHI). C-reactive protein (CRP) was measured using an Automatic Immuno-Analyzer (ADC ELISA 200; Atecom Technology).

## Sample preparation and molecular essay protocols

Blood samples (5 mL) were collected into tubes containing anticoagulant, followed by low-speed cryogenic centrifuge (TDL-5M, Michael Laboratory Instrument) at 3,000 rpm/min for 10 min. The supernatant was carefully aspirated to obtain plasma, which was then aliquoted and stored at −80 °C until analysis.

## Western blotting

Before performing Western Blot, the extracted plasma samples were thawed at 4 °C and their concentrations were determined using the Bicinchoninic Acid (BCA) protein assay. After protein denaturation, the protein was separated by 8% SDS-PAGE gel and transferred onto a PVDF membrane (IPVH00010; Millipore). The membrane was blocked with 5% skim milk for 1 h and incubated with primary antibodies, including ephrinB2 (1:1,000, #MA5-32740; Invitrogen, Waltham, MA, USA) and transferrin (1:1,000, #A19130; ABclonal, Woburn, MA, USA) at 4 °C overnight followed by secondary antibody incubation. Proteins were visualized by ECL Plus Western blotting detection reagents (RPN2132).

## Statistical analysis

Measurement data were presented as means ± standard deviation (SD) or median (interquartile range, IQR), while categorical variables were presented as patient numbers and percentages (%). Parametric t-test was used to determine whether there was a significant difference in plasma EphrinB2 expression. For multiple group data, when the data met the conditions of normal distribution and homogeneity of variance, such as age, BMI, TC, and TG, One-Way ANOVA was used for comparison. When these conditions were not met, such as for BUN, CRP, ESR, the Kruskal-Wallis H test was used for comparison. Fisher's test and Chi-square ($\chi$2) tests were used for categorical variables. The correlation analyses were performed using a Pearson correlation test. Survival analysis was conducted by the Kaplan-Meier method and Log-rank test, and patients without completed follow-up data were excluded. Logistic regression and was used to study the association between preoperative ephrinB2 levels and postoperative amputation in lower extremity PAD patients. Receiver operating characteristic (ROC) curve analysis was plotted to assess the predictive value of the area under the curve (AUC) for preoperative ephrinB2 levels in predicting postoperative amputation. Power analysis was conducted using G*Power 3.1. Other analyses were performed using IBM SPSS Statistics version 25 and GraphPad Prism 8.0 (GraphPad Software, Inc, La Jolla, CA, USA). Significance was set at $p < 0.05$.

## RESULTS

A total of 133 cases of lower extremity PAD patients were eligible for inclusion and the clinical characteristics of lower extremity PAD patients included in the study are summarized in Table 1. The mean age was 66.5 ± 12.1 years; 81.2% were either current or former smokers, and comorbidities such as hypertension (63.9%), diabetes (36.8%), and coronary artery disease (32.3%) were recorded. It was noted that there were no significant differences in drug usage across patients of different Rutherford categories.

### Elevation of plasma ephrinB2 level in lower extremity PAD patients

To evaluate the plasma ephrinB2 expression in lower extremity PAD patients, we conducted Western blot analysis to examine the samples collected. We observed a significant upregulation of ephrinB2 expression in lower extremity PAD patients compared to healthy people as control ($p < 0.001$) (Figs. 1A, 1B). Further analysis reveals a significant difference in the expression levels of ephrinB2 in plasma between Rutherford category 3 and 4 ($p = 0.023$), while no significant difference is observed between categories 4–6 (Fig. 1C).

### EphrinB2 levels positively correlate with inflammatory markers in lower extremity PAD patients

The levels of meaningful laboratory results in patients, such as white blood cell counts, neutrophil counts, and neutrophil lymphocyte ratio (NLR), show an increase with higher

**Table 1** Baseline characteristics of PAD with regard to Rutherford categories.

| Parameter | RF 3 | RF 4 | RF 5 | RF 6 | Total | *P* value |
|---|---|---|---|---|---|---|
| Patients (%) | 25 (18.8) | 46 (34.6) | 31(23.3) | 31 (23.3) | 133 (100.0) | |
| Age, years | 63.2 ± 9.6 | 67.9 ± 12.6 | 67.9 ± 10.7 | 65.8 ± 14.4 | 66.5 ± 12.1 | 0.566 |
| Female, *n* (%) | 6 (24.0) | 4 (8.7) | 7 (22.6) | 6 (19.4) | 23 (17.3) | 0.425 |
| BMI, (kg/m$^2$) | 22.6 ± 3.1 | 22.4 ± 3.4 | 21.6 ± 3.1 | 21.6 ± 3.1 | 22.1 ± 3.2 | 0.659 |
| Current smoking (%) | 21 (84.0) | 38 (82.6) | 25 (80.6) | 24 (77.4) | 108 (81.2) | 0.975 |
| Diabetes (%) | 9 (36.0) | 16 (34.8) | 10 (32.3) | 14 (45.2) | 49 (36.8) | 0.863 |
| Hypertension (%) | 17 (68.0) | 27 (58.7) | 20 (64.5) | 21 (67.4) | 85 (63.9) | 0.785 |
| Coronary heart disease (%) | 6 (24.0) | 17 (37.0) | 9 (29.0) | 11 (35.5) | 43 (32.3) | 0.820 |
| Renal insufficiency (%) | 3 (12.0) | 5 (10.9) | 11 (35.5) | 14 (45.2) | 33 (24.8) | **0.003**\* |
| WBC (×10$^9$/L) | 6.1 ± 1.9 | 7.6 ± 3.1 | 9.0 ± 3.3 | 9.1 ± 2.9 | 8.0 ± 3.1 | **<0.001**\* |
| Neutrophil (×10$^9$/L) | 3.8±1.5 | 5.2 ± 2.8 | 6.0 ± 3.2 | 6.9 ± 3.0 | 5.5 ± 2.9 | **<0.001**\* |
| Lymphocyte (×10$^9$/L) | 1.7 ± 0.3 | 1.7 ± 0.7 | 2.1 ± 0.9 | 1.5 ± 0.7 | 1.7 ± 0.7 | **0.007**\* |
| Monocyte (×10$^9$/L) | 0.4 ± 0.2 | 0.5 ± 0.2 | 0.6 ± 0.2 | 0.6 ± 0.3 | 0.5 ± 0.2 | **0.038**\* |
| Blood platelets (×10$^9$/L) | 208.0 ± 66.0 | 215.2 ± 71.0 | 293.9 ± 99.2 | 290.6 ± 118.5 | 249.8 ± 97.6 | **<0.001**\* |
| NLR | 2.2 ± 0.7 | 3.6 ± 2.4 | 3.9 ± 4.3 | 6.8 ± 5.8 | 4.2 ± 4.0 | **<0.001**\* |
| TG (mmol/L) | 1.9 ± 1.0 | 1.7 ± 1.2 | 1.4 ± 0.6 | 1.5 ± 1.0 | 1.6 ± 1.0 | 0.308 |
| TC (mmol/L) | 4.4 ± 1.1 | 4.3 ± 1.0 | 4.7 ± 0.9 | 3.9 ± 1.0 | 4.3 ± 1.1 | **0.028**\* |
| LDL-C (mmol/L) | 2.9 ± 1.0 | 2.7 ± 0.9 | 3.1 ± 0.8 | 2.4 ± 0.9 | 2.7 ± 0.9 | **0.019**\* |
| BUN (mmol/L) | 78.0 (68.8, 92.9) | 78.5 (62.3, 104.1) | 76.4 (68.1, 91.6) | 93.8 (74.6, 127.4) | 80.8 (66.9, 103.0) | **0.008**\* |
| CREA (μmol/L) | 6.2 (5.2, 7.6) | 6.4 (5.3, 8.6) | 6.4 (5.4, 7.5) | 8.5 (5.3, 11.8) | 6.4 (5.2, 8.7) | **0.020**\* |
| FBG (mmol/L) | 5.0 (4.2, 6.6) | 5.1 (4.6, 6.3) | 5.5 (4.9, 6.3) | 5.3 (4.4, 7.7) | 5.3 (4.4, 6.6) | 0.461 |
| CRP (mg/L) | 2.8 (1.8, 5.8) | 9.0 (3.8, 17.6) | 8.3 (4.9, 26.8) | 51.7 (14.7, 88.3) | 8.7 (3.6, 36.1) | **0.004**\* |
| D-Dimer (μg/ml FEU) | 0.5 (0.3, 0.6) | 0.8 (0.4, 1.7) | 0.5 (0.3, 0.9) | 0.6 (0.5, 1.6) | 0.6 (0.4, 1.2) | 0.104 |
| ESR (mm/h) | 13.0 (5.0, 15.0) | 10.0 (6.5, 27.0) | 23.0 (14.3, 49.8) | 76.0 (29.0, 88.0) | 19.0 (8.0, 59.0) | **<0.001**\* |
| PCT (ng/ml) | 0.038 (0.028, 0.049) | 0.057 (0.038, 0.114) | 0.040 (0.030, 0.065) | 0.082 (0.053, 0.424) | 0.051 (0.035, 0.084) | 0.140 |
| **Medications** | | | | | | |
| Aspirin (%) | 12 (48.0) | 17 (37.0) | 15 (48.4) | 10 (32.3) | 54 (40.6) | 0.646 |
| Statins (%) | 10 (40.0) | 17 (37.0) | 15 (48.4) | 12 (38.7) | 54 (40.6) | 0.897 |
| β-blocker (%) | 11 (44.0) | 28 (60.9) | 14 (45.2) | 12 (38.7) | 65 (48.9) | 0.362 |
| ACEI/ARB (%) | 6 (24.0) | 18 (39.1) | 15 (48.4) | 11 (35.5) | 50 (37.6) | 0.461 |

**Notes:**
RF, Rutherford category; WBC, White Blood Cell; NLR, Neutrophil-to-Lymphocyte Ratio; NMR, Neutrophil-to-Monocyte Ratio; TG, Triglyceride; TC, Total Cholesterol; LDL-C, Low-density Lipoprotein Cholesterol; BUN, Blood Urea Nitrogen; CREA, Creatinine; FBG, Fasting blood glucose; CRP, C reactive protein; ESR, Erythrocyte Sedimentation Rate; PCT, Procalcitonin. Comorbidities were defined on the basis of given ICD-10 codes. Values represent mean ± SD or median (lower quartile, upper quartile), or *n* (%) as indicated. Significant *p* values were presented in bold and marked with an asterisk (\*).

Rutherford categories ($p < 0.001$) (Table 1). We conducted a correlation analysis between plasma EphrinB2 levels and inflammatory markers, and the results demonstrated a positive correlation between EphrinB2 and white blood cell counts (r = 0.204, p = 0.018), neutrophil counts (r = 0.174, p = 0.045), as well as NLR (r = 0.223, p = 0.009) (Figs. 1D, 1E and 1G).

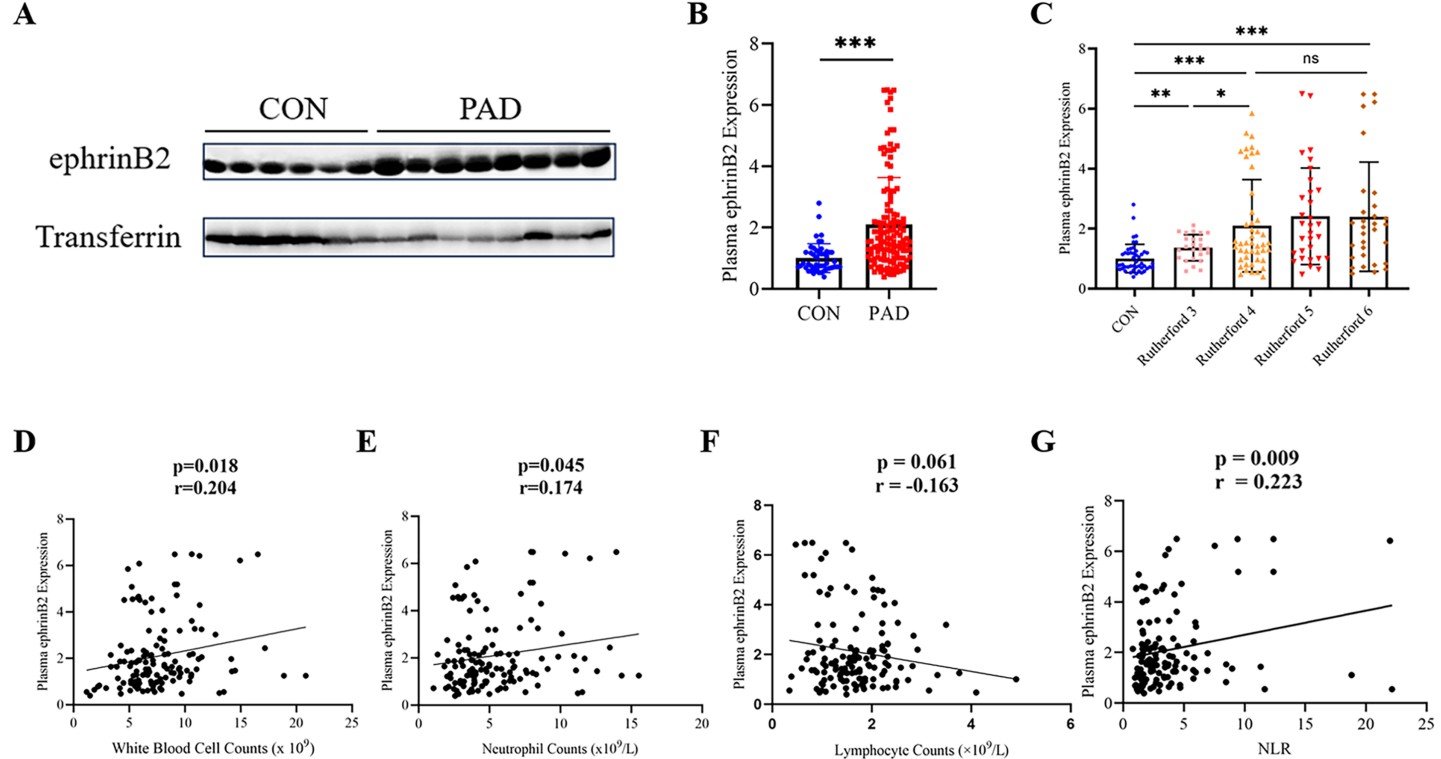

**Figure 1 Plasma ephrinB2 levels are elevated in lower extremity PAD patients and positively correlate with inflammatory markers.** (A–C) Western blotting analysis demonstrated elevated expression of plasma ephrinB2 in lower extremity PAD patients ($p < 0.001$), with significant differences observed between Rutherford categories 3 and 4 ($p = 0.023$). Linear relationships between plasma ephrinB2 level and (D) White blood cell counts, (E) neutrophil counts, (F) lymphocyte counts and (G) neutrophil-to-lymphocyte ratio (NLR). ns: $p \geq 0.05$; $^{*}p < 0.05$; $^{**}p < 0.01$; $^{***}p < 0.001$.

## High levels of plasma ephrinB2 indicate a higher risk of postoperative amputation events

During the 30-month follow-up, 11 individuals had amputations and 14 individuals died. Among the deceased, six died from cardiac pathologies, five from COVID-19, two from cancer, and one from sepsis.

We investigated the predictive roles of ephrinB2 and its combination with NLR in this study. The ROC analysis revealed that plasma ephrinB2 levels and NLR have potential predictive value for amputation events within 30 months after lower extremity PAD endovascular revascularization, with AUCs of 0.659 ($p = 0.047$; 95% confidence interval (CI) [0.502~0.861]) and 0.730 ($p = 0.012$; 95% CI [0.559~0.900]), respectively. When the test results for ephrinB2 and NLR were considered jointly, the ROC analysis revealed an AUC of 0.811 ($p < 0.001$; 95% CI [0.675–0.918]), indicating higher predictive capability (Fig. 2A, Table 2). However, when ephrinB2 is applied individually, it did not demonstrate a predictive role in postoperative 30-month mortality events ($p = 0.153$, 95% CI [0.383–0.714]) (Fig. 2B, Table 2). To further study the relationship between plasma ephrin levels and amputation events within 30 months after revascularization, as shown in

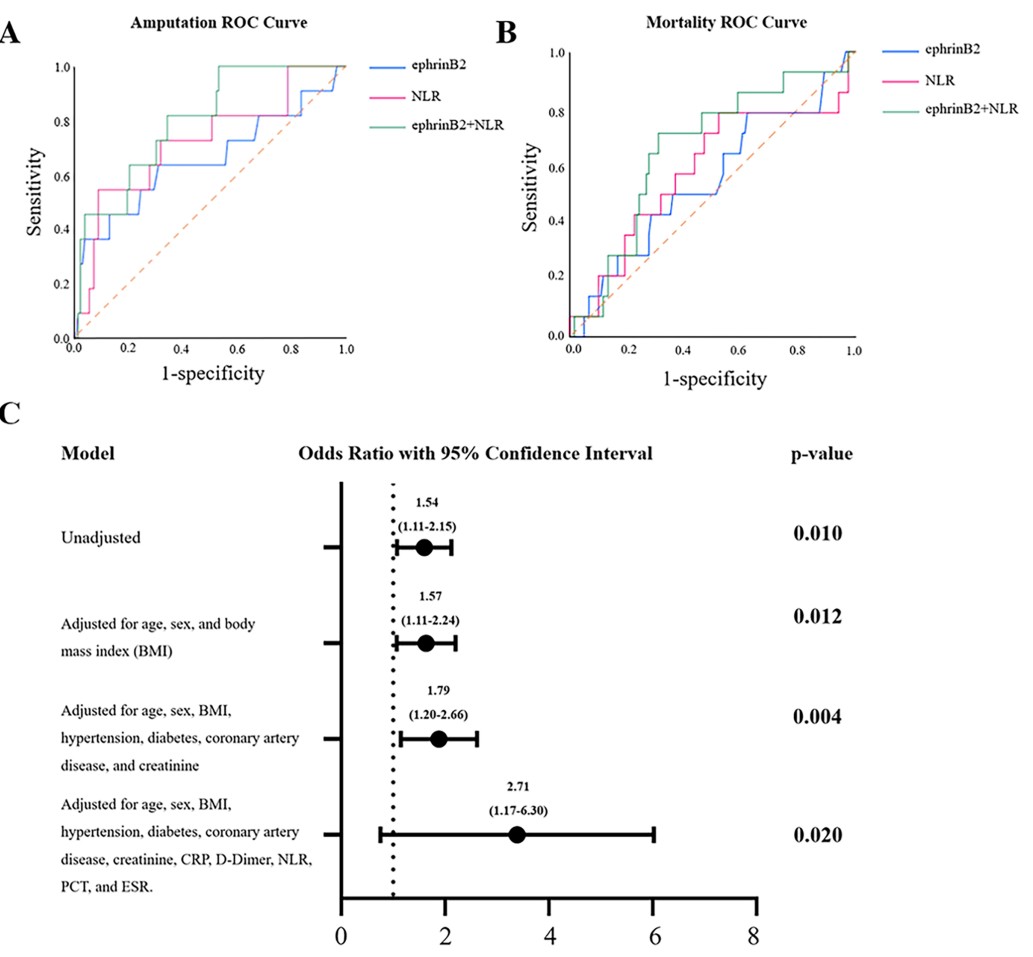

**Figure 2 Predictive analysis in post-revascularization outcomes.** (A) ROC curve analysis of plasma ephrinB2 level, NLR, and their combination in predicting amputation after revascularization. (B) ROC curve analysis of plasma ephrinB2 level, NLR, and their combination in predicting mortality after revascularization. (C) Logistics regression forest map of plasma ephrinB2 level and adjusted factors in post-vascular revascularization amputation.

Fig. 2C, we conducted a logistic analysis. The results indicated a significant correlation between ephrinB2 expression and the 30-month post-amputation rate. Furthermore, even after adjusting for age, gender, and other clinical and laboratory parameters, the association remained statistically significant.

Depending on the optimal cut-off value according to ROC for the 30 months amputation obtained from Youden's index (2.13 for plasma ephrinB2 level), the outcomes were further analyzed after dividing the patients into paired groups: low-ephrinB2/high-ephrinB2. As shown in Table 3, in the high-ephrinB2 group, there was a higher incidence of amputation events within 30 months after endovascular revascularization, while there was no difference in mortality events.

The Kaplan-Meier plot for the 30 months amputation and mortality based on the optimal cut-off value of plasma ephrinB2 level for all patients is shown in Fig. 3. The results indicated that the high ephrinB2 group had a higher amputation risk ($p = 0.048$).

**Table 2 ROC analysis of predictive indicators for amputation and mortality within 30 months after vascular reconstruction.**

| Variables | AUC | 95% Confidence interval | *P* value | Cutoff | Sensitivity | Specificity |
|---|---|---|---|---|---|---|
| **Amputation** | | | | | | |
| EphrinB2 | 0.659 | [0.502–0.861] | **0.047** | 2.13 | 63.6 | 68.9 |
| NLR | 0.730 | [0.559–0.900] | **0.012** | 7.3 | 54.5 | 91.0 |
| EphrinB2+NLR | 0.811 | [0.675–0.918] | **<0.001** | | 81.8 | 89.6 |
| **Mortality** | | | | | | |
| EphrinB2 | 0.549 | [0.383–0.714] | 0.153 | 1.29 | 78.6 | 37.8 |
| NLR | 0.584 | [0.305–0.410] | 0.105 | 2.57 | 78.6 | 47.9 |
| EphrinB2+NLR | 0.661 | [0.515–0.806] | **0.049** | | 71.4 | 68.9 |

Note:
NLR, neutrophil lymphocyte ratio; Significant *P*-values were presented in bold.

**Table 3 Univariate analysis of plasma ephrinB2 level and all adverse event occurrences during the study period for all patients.**

| EphrinB2 = 2.13 | Amputation | Mortality |
|---|---|---|
| LOW-ephrinB2 *vs.* High-ephrinB2 | 4/88 (4.54%) *vs.* 7/45 (15.56%) | 8/88 (9.09%) *vs.* 6/45 (13.33%) |
| | *p* = 0.0438 | *p* = 0.5522 |
| | OR: 3.868; | OR: 1.538; |
| | 95% CI [1.115–12.25] | 95% CI [0.480–5.030] |

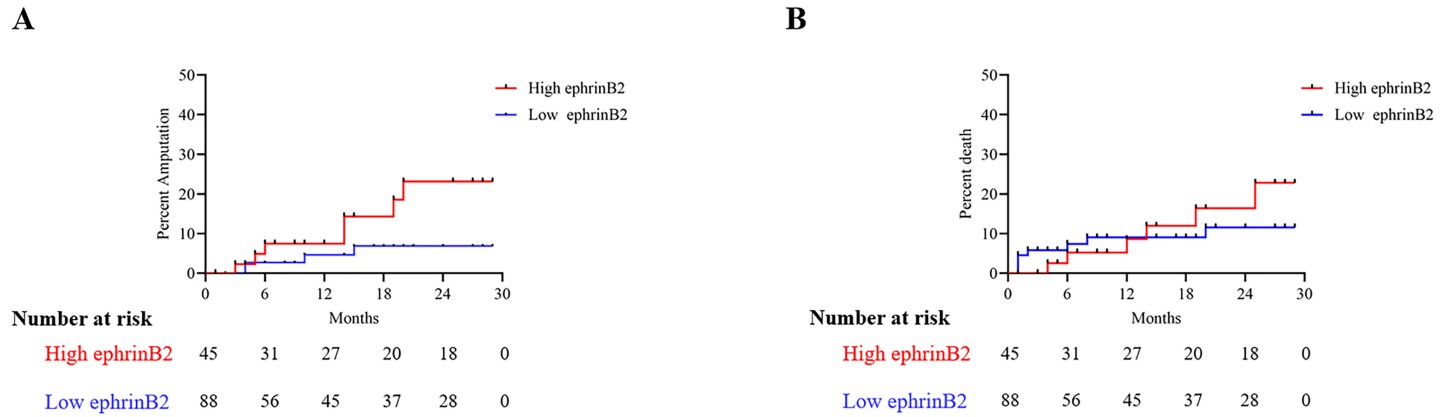

**Figure 3 Kaplan-meier survival curve for 30-month survival and mortality rates.** Kaplan–Meier curves showing (A) amputation probability according to ephrinB2 optimal cut-off value (*p* = 0.044; log-rank *p*), (B) mortality probability according to ephrinB2 optimal cut-off value (*p* = 0.551; log-rank *p*).

## DISCUSSION

In this study, we investigated the relationship between plasma ephrinB2 levels and lower extremity PAD patients. Our findings revealed that patients with lower extremity PAD had significantly higher plasma ephrinB2 levels compared to the healthy control group. Additionally, correlation analysis demonstrated a positive relationship between plasma ephrinB2 levels and some hematological markers including white blood cell counts,

neutrophil counts, as well as NLR. Logistic regression analysis suggested that higher preoperative plasma ephrinB2 levels might be indicative of an increased probability of amputation within 30 months after endovascular revascularization. Moreover, ROC curve analysis indicated that the combined use of plasma ephrinB2 levels and NLR showed higher predictive accuracy (AUC = 0.811) for amputation events within this timeframe, compared to NLR alone (AUC = 0.730). Also, the ephrinB2-NLR combination is higher in specificity and sensitivity to predict amputation risk compare to ephrinB2 or NLR alone.

PAD not only has a negative impact on the health and quality of life of patients but also imposes a significant economic burden on society (*Criqui & Aboyans, 2015*). Numerous studies have indicated a close association between the occurrence and development of PAD and the systemic inflammatory response (*Aday & Matsushita, 2021*; *Brevetti et al., 2010*; *Fort-Gallifa et al., 2016*). Previous studies have shown that during inflammation process, the expression of ephrinB2 in capillaries is upregulated. Activating ephrinB2 ligands can lead to a decrease in the integrity of endothelial cell junctions and enhance the pro-inflammatory phenotype of the endothelium (*Liu et al., 2014*; *Sweet et al., 2013*). Furthermore, under conditions of pre-atherosclerotic blood flow, ephrinB2 is upregulated and functions as a chemoattractant, facilitating white blood cell migration even in the absence of other chemotactic factors (*Cao et al., 2018*). ADAM10 is a cell surface shedding enzyme that regulates physiological processes, including Notch signaling (*Tsai et al., 2014*). Despite some studies suggesting that ephrinB2 can be cleaved by the enzyme Adam10 to release active proteins in specific circumstances (*Lagares et al., 2017*; *Mueller et al., 2021*), the connection between plasma-free ephrinB2 levels and the activation of inflammatory cells remains unclear. Nevertheless, this connection may play a role in the pathogenesis of peripheral arterial disease. Lower extremity PAD often leads to ischemia and necrosis in the affected endovascular supply area, with endovascular revascularization being a common treatment option. However, despite undergoing revascularization, some patients may still require amputation due to already gangrene tissue or deterioration later. Therefore, predicting amputation after vascularization is important for physicians and patients. *Gary et al. (2013)* conducted a study involving 2,121 lower extremity PAD patients and found that an elevated NLR is significantly associated with an increased risk of critical limb ischemia (CLI). *Shin et al. (2017)* conducted a study on 381 patients with acute myocardial infarction (AMI) who underwent percutaneous coronary intervention (PCI), which showed that elevated NLR and CRP levels after PCI in AMI patients are associated with an increased risk of long-term mortality, suggesting that the combined use of both markers enhances the predictive value for long-term prognosis. In our study, we observed a significant increase in NLR with disease severity. Therefore, we chose NLR as an additional variable and used ROC curve analysis to assess its predictive capabilities when used alone and in combination with ephrinB2 for amputation events. We found that the combined application of plasma ephrinB2 levels and NLR demonstrated higher predictive accuracy for amputation events both in sensitivity and specificity, providing new insights into the role of plasma ephrinB2 in lower extremity PAD.

Membrane-bound ephrinB2 and its receptor EphB4 play a crucial role in blood and lymphatic vessel formation under both normal and pathological conditions (*Abéngozar*

*et al., 2012*; *Masumura et al., 2009*), and they are indispensable for arterial remodeling. Our results revealed that elevated levels of free ephrinB2 in plasma may indicate the development of lower extremity PAD or worse postoperative outcomes following vascularization. This phenomenon may be attributed to ephrinB2 related molecular signaling transduction in mechanism that led to the cleavage of ephrinB2 by ADAM10 on endothelial cells, resulting in reduced endothelial cell stability and hindered neovascularization. Additionally, free ephrinB2 may also participate in the inflammatory response. Therefore, further in-depth mechanism study is needed to elucidate the role of ephrinB2 in lower extremity PAD.

This study has several limitations. Firstly, the study design is cross-sectional, which only allows for correlations between variables and does not establish causal relationships. Secondly, the study is a single-center investigation with a relatively small sample size, and the participants were exclusively recruited from specific hospitals and underwent rigorous screening. This led to a relatively small and varied number of patients in each Rutherford grade, which may affect the accuracy of our results. Additionally, due to the profound impact of the COVID-19 pandemic on postoperative survival outcomes during the study period, it may not accurately reflect the predictive role of plasma ephrinB2 under normal circumstances. Therefore, further research on the association between plasma ephrinB2 levels and postoperative patient mortality is still needed.

## CONCLUSIONS

In summary, we revealed elevated levels of plasma ephrinB2 in lower extremity PAD patients and explored its potential association with atherosclerotic inflammation. This study investigated the predictive capability of ephrinB2, both independently and in combination with NLR, for forecasting amputation events occurring within 30 months following lower extremity PAD endovascular revascularization. This study may offer a novel approach for predicting post-PAD amputation events and broaden the scope of research on the significance of ephrinB2 molecule in lower extremity PAD.

### Funding
This research was supported by the National Natural Science Foundation of China (NSFC) (Grant Number 82120108005) and the Central South University Research Program of Advanced Interdisciplinary Studies (Grant No. 2023QYJC024). The funders had no role in study design, data collection and analysis, decision to publish, or preparation of the manuscript.

### Grant Disclosures
The following grant information was disclosed by the authors:
National Natural Science Foundation of China (NSFC): 82120108005.
Central South University Research Program of Advanced Interdisciplinary Studies: 2023QYJC024.

## Competing Interests

The authors declare that they have no competing interests.

## Author Contributions

- Pengcheng Guo conceived and designed the experiments, performed the experiments, analyzed the data, prepared figures and/or tables, and approved the final draft.
- Lei Chen conceived and designed the experiments, prepared figures and/or tables, authored or reviewed drafts of the article, and approved the final draft.
- Dafeng Yang conceived and designed the experiments, analyzed the data, authored or reviewed drafts of the article, and approved the final draft.
- Lei Zhang analyzed the data, prepared figures and/or tables, authored or reviewed drafts of the article, and approved the final draft.
- Chang Shu performed the experiments, prepared figures and/or tables, and approved the final draft.
- Huande Li conceived and designed the experiments, performed the experiments, prepared figures and/or tables, and approved the final draft.
- Jieting Zhu performed the experiments, authored or reviewed drafts of the article, and approved the final draft.
- Jienan Zhou analyzed the data, authored or reviewed drafts of the article, and approved the final draft.
- Xin Li conceived and designed the experiments, analyzed the data, authored or reviewed drafts of the article, and approved the final draft.

## Human Ethics

The following information was supplied relating to ethical approvals (*i.e.*, approving body and any reference numbers):

This study was approved by the Ethics Committee of The Second Xiangya Hospital of Central South University (No. 2022-Clinical Research-113).

## Data Availability

The raw measurements are available in the Supplemental Files.

## Supplemental Information

Supplemental information for this article can be found online at http://dx.doi.org/10.7717/peerj.17531#supplemental-information.

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
