# Peer review of "Predictive value of plasma ephrinB2 levels for amputation risk following endovascular revascularization in peripheral artery disease"

_PeerJ, doi:10.7717/peerj.17531_

## Round 0.1 · original submission · Major Revisions

It was quite difficult to find expert reviewers in this field. Please consider the opinions of valuable reviewers carefully.

Reviewer 1 ·

Basic reporting

no commet

Experimental design

Researchers stated that in the study in question, they collected 133 blood samples from patients diagnosed with lower extremity PAH and 51 blood samples from healthy volunteer donors. However, the fact that they did not perform a power analysis when determining these numbers was identified as a deficiency. The authors' failure to conduct this analysis may undermine the statistical reliability of the study and limit the validity of the results obtained. Power analysis allows reaching more robust and reliable results by evaluating the power of sample size, interaction level and other factors in statistical analysis. Therefore, the lack of relevant statistical power analysis indicates that the study is methodologically incomplete and caution should be exercised in interpreting the findings.

Validity of the findings

Researchers stated that serum ephrinB2 levels were significantly higher in patients compared to the control group. Additionally, a positive correlation was found between plasma EphrinB2 levels and white blood cell count, neutrophil count and neutrophil-lymphocyte ratio. In addition, it has been stated that plasma ephrinB2 level, NLR and the combination of these factors are evaluated as determinants of amputation events within 30 months after endovascular revascularization. In these evaluations, AUC values were reported. Finally, it was stated that amputation events were more common within 30 months after endovascular revascularization in the group with high ephrinB2 levels. These findings indicate the potential importance of serum ephrinB2 levels, NLR, and the combination of these parameters in predicting amputation events.

Reviewer 2 ·

Basic reporting

Dear Editor/Authors;
Thank you for your interest. The authors have evaluated “Predictive value ofplasma ephrinB2 levels for amputation risk following endovascular revascularization in peripheral artery disease”. The authors should make some revisions to the following recommendations.

The manuscript is written under the general formats/structure and literature and will make a scientific contribution.

Experimental design

• The methods of all analyzes performed in the study (blood routine, inflammation markers, etc.) and the devices used for analysis should be provided.
• The sample extraction methods in Western Blotting analyses should be specified. Which extraction method were used?
• The housekeeping protein and its brand in Western blotting analysis should be specified.
• It should be stated which data were analyzed by the Mann-Whitney U test and which values were analyzed by the T-test, and it should be explained why both parametric and non-parametric tests were used.
• Why were NLR and ephrin 2 not evaluated along with CRP level in the study?
• Although inflammation is an important condition in PAD patients, no pro-inflammatory cytokines were analyzed. The pro-inflmmatory cytokines analyzes will make a significant contribution to the quality of the research.

Validity of the findings

no comment

---

## Round 0.2 · accepted · Accept

The study, with its revised version made as a result of reviewer opinions, brings a meaningful innovation to the existing literature and increases knowledge on the subject. The research questions were stated clearly, and the methods used were applied appropriately and meticulously.

Reviewer 1 ·

Basic reporting

no comment

Experimental design

no comment

Validity of the findings

no comment

Additional comments

no comment

Reviewer 2 ·

Basic reporting

I thank the authors for their responses and edits.

Experimental design

no comment

Validity of the findings

no comment

Additional comments

I thank the authors for their responses and edits.